# Deformation Anomalies Accompanying Tsunami Origination

Grigory Dolgikh [1,2,*] and Stanislav Dolgikh [1,2,*]

1   V.I. Il'ichev Pacific Oceanological Institute, Far Eastern Branch Russian Academy of Sciences,
    690041 Vladivostok, Russia
2   AEROCOSMOS Research Institute for Aerospace Monitoring, 105064 Moscow, Russia
*   Correspondence: dolgikh@poi.dvo.ru (G.D.); sdolgikh@poi.dvo.ru (S.D.)

**Abstract:** Basing on the analysis of data on variations of deformations in the Earth's crust, which were obtained with a laser strainmeter, we found that deformation anomalies (deformation jumps) occurred at the time of tsunami generation. Deformation jumps recorded by the laser strainmeter were apparently caused by bottom displacements, leading to tsunami formation. According to the data for the many recorded tsunamigenic earthquakes, we calculated the damping ratios of the identified deformation anomalies for three regions of the planet. We proved the obtained experimental results by applying the sine-Gordon equation, the one-kink and two-kink solutions of which allowed us to describe the observed deformation anomalies. We also formulated the direction of a theoretical deformation jump occurrence—a kink (bore)—during an underwater landslide causing a tsunami.

**Keywords:** earthquake; tsunami; laser strainmeter; deformation jump; sine-Gordon equation; kink; anti-kink; underwater landslide

## 1. Introduction

We know that tsunamis are some of the most dangerous and catastrophic phenomena on Earth, which cause significant damage to humanity. A typical example is the tsunami that hit the Indian Ocean on 26 December 2004, killing more than 283,000 people. This was caused by a powerful earthquake with a maximum magnitude of about 9.3 [1]. Tsunamis affect various regions of the planet, although this is especially true for Japan. Considering the extent to which Japan suffers from the impacts of earthquakes and tsunamis and its high level of scientific and technical development, we can expect that more advanced scientific and technical ideas aimed at predicting the occurrence and development of earthquakes and tsunamis will be concentrated in this region of the planet. While the short-term forecasting of earthquakes is far from being solved, the detection of the moments of tsunami origination seems to be quite solvable. The Japanese Islands and the adjoining water areas are "crammed" with various seismic stations, GPS receivers, bottom seismic stations, and high-precision sea and ocean level meters. Nevertheless, the events of 2011 "exposed" the short-term tsunami forecasting problems even more.

Presently, the traditional method of short-term tsunami forecasting is based on seismological information (earthquake magnitude, main shock time, and epicenter location) [2]. An earthquake magnitude that exceeds a predetermined threshold, which will be different for different tsunamigenic zones, usually results in a tsunami warning. This approach, based on the "magnitude and geographical principle", is simple; it helps reduce the number of missed tsunamis, but also gives false alarms. Most current early tsunami warning systems are based on seismic data.

For example, the Pacific Tsunami Warning Center (PTWC) uses seismic data in conjunction with long-period wave (W-phase) data for global tsunami warnings in the Pacific [3]. Another tsunami warning center is the Japan Meteorological Agency (JMA), which provides local tsunami warnings within up to 3 min after near-field earthquakes by analyzing seismic data [4], then updates the warnings using the forms of seismic wave and tsunami

data [5]. The W-phase appears in seismic recordings between P-waves and S-waves and can be used to estimate the seismic moment, epicenter location, and fault mechanism. The efficiency of W-phase inversion has already been demonstrated in many studies and is actively used by tsunami warning centers [6,7].

Recently, two independent approaches have been proposed to determine the energy of a tsunami source: one is based on Deep-Ocean Assessment and Reporting of Tsunamis (DART) data during tsunami propagation, and the other on ground-based coastal global positioning system (GPS) data during tsunami generation. The GPS approach takes into account the dynamic earthquake process, while the DART inversion approach provides an actual estimate of the tsunami energy and propagating tsunami waves. Both approaches lead to coordinated energy scales for the earlier-studied tsunamis. Inspired by these promising results, the authors [8] researched an approach to determine the tsunami source energy in real time by combining the two methods. At the first stage, the tsunami source is determined immediately after the earthquake using the global GPS network for early warnings in the near-field zone. Then, the tsunami energy is defined more precisely based on the nearest DART measurements to improve the accuracy of forecasts or cancel the alarm. The combination of these two real-time networks could offer an attractive opportunity for the early detection of tsunami threats to save more lives and for the early cancellation of tsunami warnings to avoid unnecessary false alarms. In the past decade, the number of open ocean sensors capable of analyzing information about a passing tsunami has steadily increased, especially thanks to the national cable networks and international efforts such as DART systems. The received information is analyzed in order to warn people about a tsunami. Most of the current warnings, which include tsunamis, are aimed at the mid-to-far zone regions. In [8], the main directions of research using DART and GPS systems were formulated; however, the failure of this system in the Indian Ocean, when a powerful tsunami was missed, plunged once optimistic scientists into despondency. For early tsunami warnings, various methods can be used, including methods based on space monitoring of earthquake-endangered areas [9–11].

We should note that tsunamis can be caused not only by earthquakes, but also by underwater landslides, volcanic activity, and simply by the collapse of mountain massifs into the sea [12].

For the first time period, a deformation anomaly was recorded during the registration of a tsunamigenic earthquake, the laser strainmeter of which showed the form of a deformation jump that occurred after the earthquake started [13]. Later, this result was generalized in [14] and the deformation methods used for determining the tsunamigenic nature of earthquakes were based on it. The potential of this method is associated with the fact that its development will allow the nature of the movement of the Earth's crust to be determined remotely at planetary distances, which sets in motion huge masses of water that degenerate into tsunamis during their development. It is clear that oscillations originating in the source of an earthquake do not cause a tsunami. These oscillations are associated with the parameters of continuity breaches, i.e., with their geometrical dimensions and elastic plate deformations. As a rule, these oscillations occur in the time range from the first minutes to the first ten seconds. These oscillations will never cause a tsunami. Only quick displacements of huge masses of the Earth's crust, which unfortunately are not recorded by any broadband seismographs, lead to a tsunami. In this paper, we will consider the peculiarities of the appearance of the deformation anomalies accompanying (attending) tsunamis, using several examples and with descriptions of the physical mechanisms of their occurrence and development. The development prospects in this direction are associated with the fact that the speed of such deformation anomalies is more than an order of magnitude higher than the speed of tsunami propagation, which is extremely important for warning services.

## 2. Recording Complex

The deformation anomaly, described in [13] and to a greater extent corresponding to the concept of a "deformation jump", showed a small amplitude equal to about 60 μm and was recorded by a laser strainmeter at a distance of about 5600 km from its location of origin. It is clear that such deformation anomalies cannot be recorded by any broadband seismograph (velocimeter, accelerometer, etc.), since these instruments are unable to register such disturbances. We could talk about the prospects of using GPS receivers to register such anomalies; however, at such distances from the locations of deformation anomalies, their registration by GPS receivers is impossible because the main samples of GPS receivers can only provide displacement registration accuracy of about 1 mm, which is much larger than the values of the registered anomalies. In the Conclusions section of this paper, we will discuss where these GPS receivers can be used. Currently, to register deformation anomalies associated with the process of tsunami generation, the most efficient instruments are laser strainmeters, which are capable of measuring microdisplacements of the Earth's crust in the frequency range of 0 (conditionally) to 1000 Hz with high accuracy (up to 1 pm) [15]. In our study, we will use the data on variations in the microdisplacements of the Earth's crust obtained from the horizontal laser strainmeter, which were created on the base of a Michelson interferometer with an unequal measuring arm length of 52.5 m and a "north–south" orientation. This instrument uses a frequency-stabilized helium–neon laser manufactured by Melles–Griott with long-term stability of $10^{-9}$ as a light source [16]. Recently, we modernized this laser strainmeter, providing it with a frequency-stabilized laser, stabilized along the iodine lines in the eleventh digit, and an improved recording system. After the modernization process, it could detect variations of microdisplacements in the Earth's crust in the frequency range of 0 (conditionally) to 1000 Hz, with accuracy of 0.03 nm. The laser strainmeter was installed in thermally insulated underground rooms at depths of about 3–5 m under the Earth's surface at 42°34.798′ N, 131°9.400′ E, at an altitude of about 60 m above sea level. Figure 1 shows a general view of the underground part of the central interference unit (left) and the underground beam guide (right). Experimental data are transmitted via a cable line to the recording computer, where after pre-processing the data files are formed, with a sampling frequency of 1 kHz and duration of 1 h. The laser strainmeter is a part of the seismoacoustic and hydrophysical complex, located in the south of the Primorsky Territory of Russia at the sea hydrophysical study site of POI FEB RAS "Shultz Cape" [17]. The main purpose of the complex is studying the nature of variations in microdeformations of the Earth's crust; fluctuations in atmospheric and hydrospheric pressure over wide-frequency and dynamic ranges; and the regularities of the emergence, development, and transformation of oscillations and waves of the sonic and infrasonic ranges. The measurements at the complex are carried out in continuous mode and all obtained data are input into the database (approximately 10 terabyte capacity), which is being steadily complemented. A precise time clock based on the Trimble 5700 GPS instrument is used for synchronization.

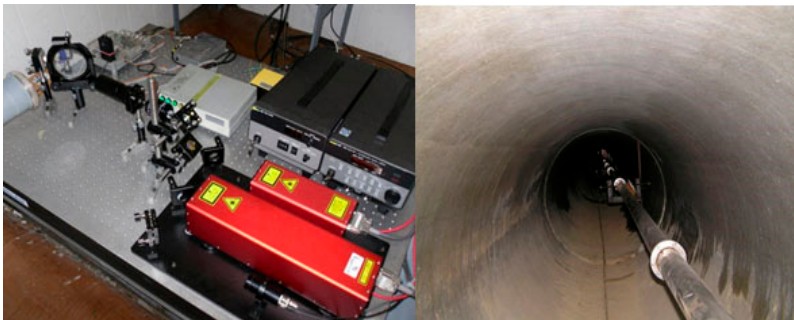

**Figure 1.** Modernized 52.5 m laser strainmeter. The central interference unit with a frequency-stabilized helium–neon laser, which is stabilized along the iodine lines (frequency stability $10^{-11}$, (**left**)), as well as an underground beam guide (**right**).

## 3. Registration of Tsunamigenic Earthquakes Deformation Anomalies

The laser strainmeter, which has been operating since 2000, has recorded many tsunamigenic and non-tsunamigenic earthquakes. Let us consider the specific differences in the behaviors of tsunamigenic and non-tsunamigenic earthquakes using the two example earthquakes described below. The first powerful tsunamigenic earthquake occurred on 26 December 2004, while the second non-tsunamigenic earthquake occurred on 4 August 2000. The epicenter of the first tsunamigenic earthquake was located at 3.30° N, 95.87° E, about 160 km west of Sumatra, at a depth of 30 km below the sea level. The distance from the earthquake epicenter to the laser strainmeter location was approximately 5600 km. The laser strainmeter recording of a peculiar signal from a tsunamigenic earthquake is shown in Figure 2. The recording shows a powerful deformation anomaly that appeared a short time after the earthquake started, with an amplitude of about 59.3 μm. The amplitude of this anomaly was much greater than the amplitude of the daily tide, as observed at the instrument location. In Figure 2, the earthquake onset is marked with an arrow. The laser strainmeter recorded the tsunamigenic earthquake signal at 19 min 54 s after the earthquake started.

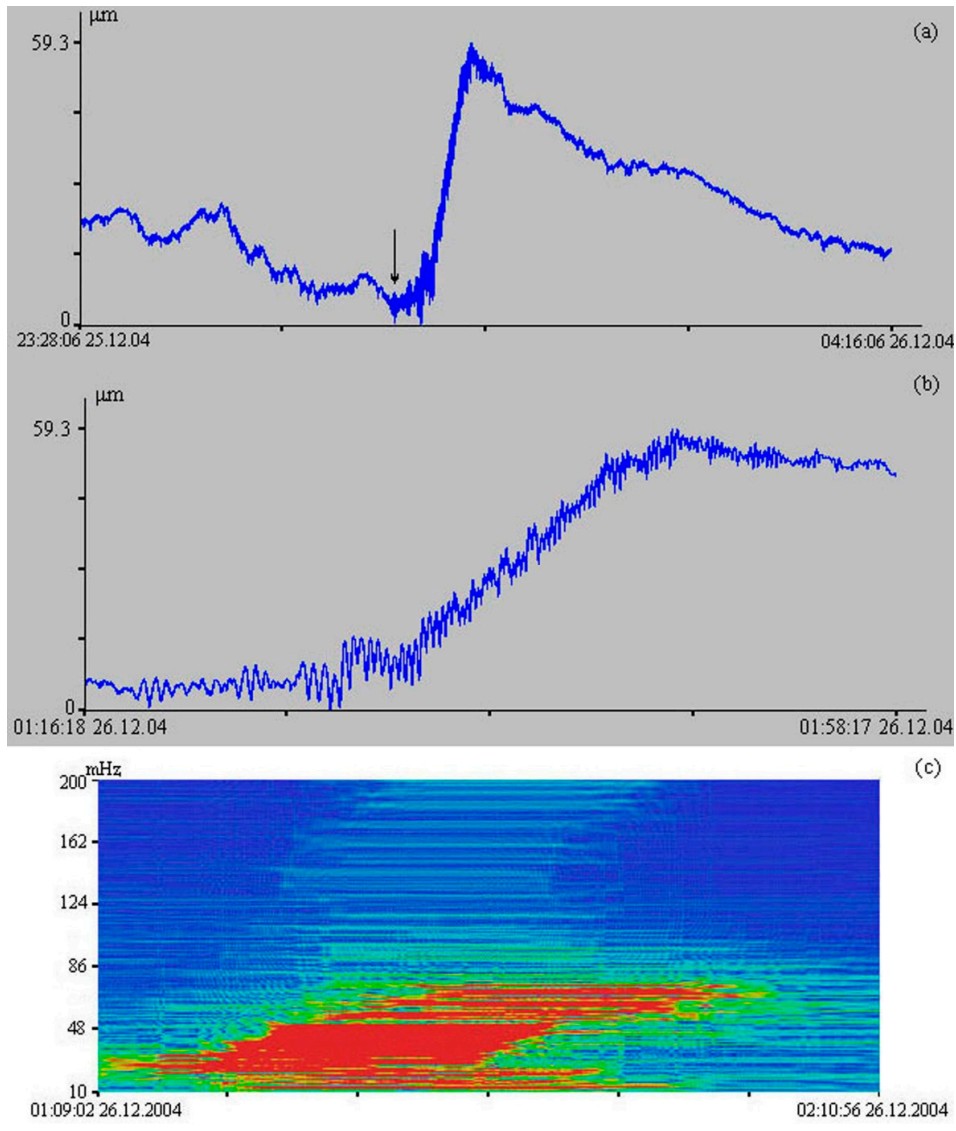

**Figure 2.** Fragment of the 52.5 m laser strainmeter recording from December 2004 (**a**), enlarged fragment of the laser strainmeter recording (**b**), and dynamic spectrogram of the laser strainmeter recording (**c**). Universal time is shown on the abscissa axis.

Analyzing the dynamic spectrogram shown in Figure 2c, we found that the periods of the main oscillations caused by the earthquake gradually decreased from 30 to 14 s. If we know the relation connecting the propagation speed of the elastic waves with the period of oscillations, the magnitude of the change in the period of the main oscillations, and the time during which this change occurred, we can determine the distance to the earthquake site. The dynamic spectrogram also shows a strong disturbance in the low-frequency range.

We must note that the results from processing the space monitoring data, which was carried out after this catastrophic tsunamigenic earthquake, showed that ionospheric anomalies were observed 4–5 days before it, which were registered by way of analyzing data from the GPS satellite navigation system. These anomalies manifested themselves as changes in the electron density profiles [11] and in changes in the total electron content (TEC) in the ionosphere in the Sumatra area [10]. At the same time, the use of a geomechanical model [18] for this region showed that the recorded increase in atmospheric pressure led to an increase in the stress–strain state of the Earth's crust and brought it closer to the strength limit before the Sumatra earthquake [10]. As an example of a non-tsunamigenic earthquake, let us consider the 52.5 m laser strainmeter recording from August 2000 (Figure 3, top). At that time there was a recording of the earthquake, which occurred on 4 August 2000 at 21:13:05 (hereinafter—universal time) at N48.85° and E142.42°, at a depth of 33 km, with a magnitude of 7.1. The recording showed oscillations of about 16 s, which were specific to an earthquake, although there was no deformation jump. The dynamic spectrogram in Figure 3 (bottom) shows that the amplitudes of oscillations in the period range of about 16 s are much greater than the amplitudes of oscillations in the low-frequency range.

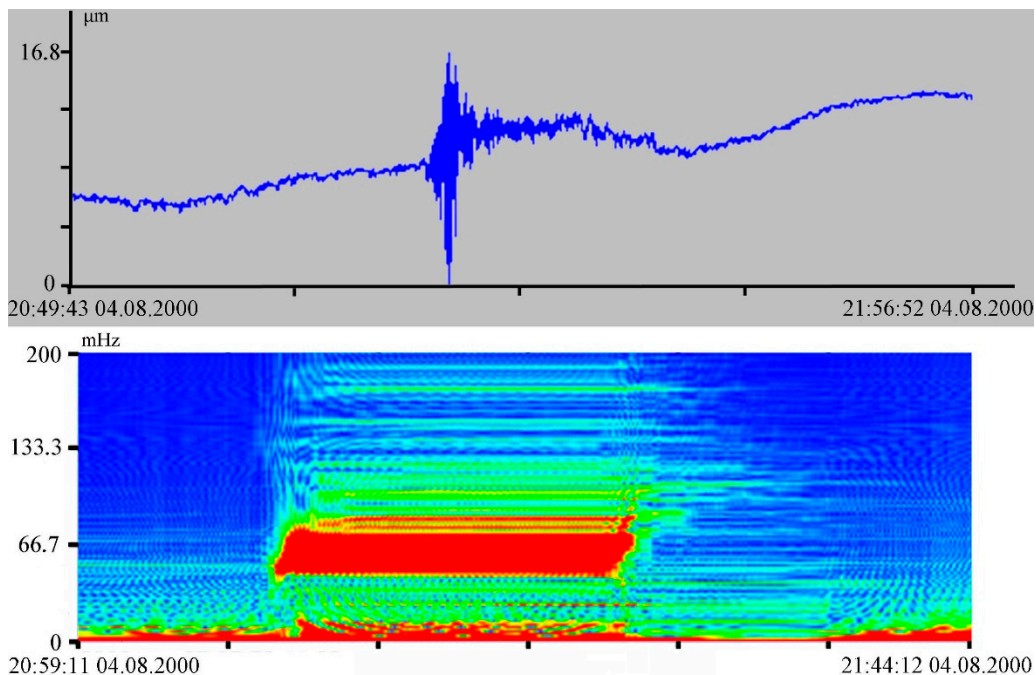

**Figure 3.** Fragment of the 52.5 m laser strainmeter recording from August 2000 (**top**) and a dynamic spectrogram of the laser strainmeter recording (**bottom**).

Analysis of the laser strainmeter records of tsunamigenic and non-tsunamigenic earthquakes showed that the deformation jump recording was specific to a tsunamigenic earthquake only.

Further, we will highlight some of the peculiarities of the appearance and development of deformation anomalies at the times of tsunami generation in the three tsunami-endangered regions, namely Indonesia, Chile, and the west coast of North America.

### 3.1. Earthquakes in Indonesia

The first powerful earthquake occurred on 11 April 2012 at 08:38:36 on the west coast of northern Sumatra, Indonesia, at 2.327° N, 93.063° E, at a depth of 20 km and a magnitude of 8.6. The maximum recorded tsunami wave height was 1.08 m. The distance from the earthquake epicenter to the location of the laser strainmeter was more than 5800 km. The laser strainmeter recorded this earthquake signal almost 18 min later at 08:55:39. The average propagation speed of the elastic wave was 5.66 km/s. In the dynamic spectrogram (Figure 4a), we can identify oscillations with periods ranging from 30 to 14 s that are peculiar to an earthquake. There is also a strong disturbance in the lower frequency range.

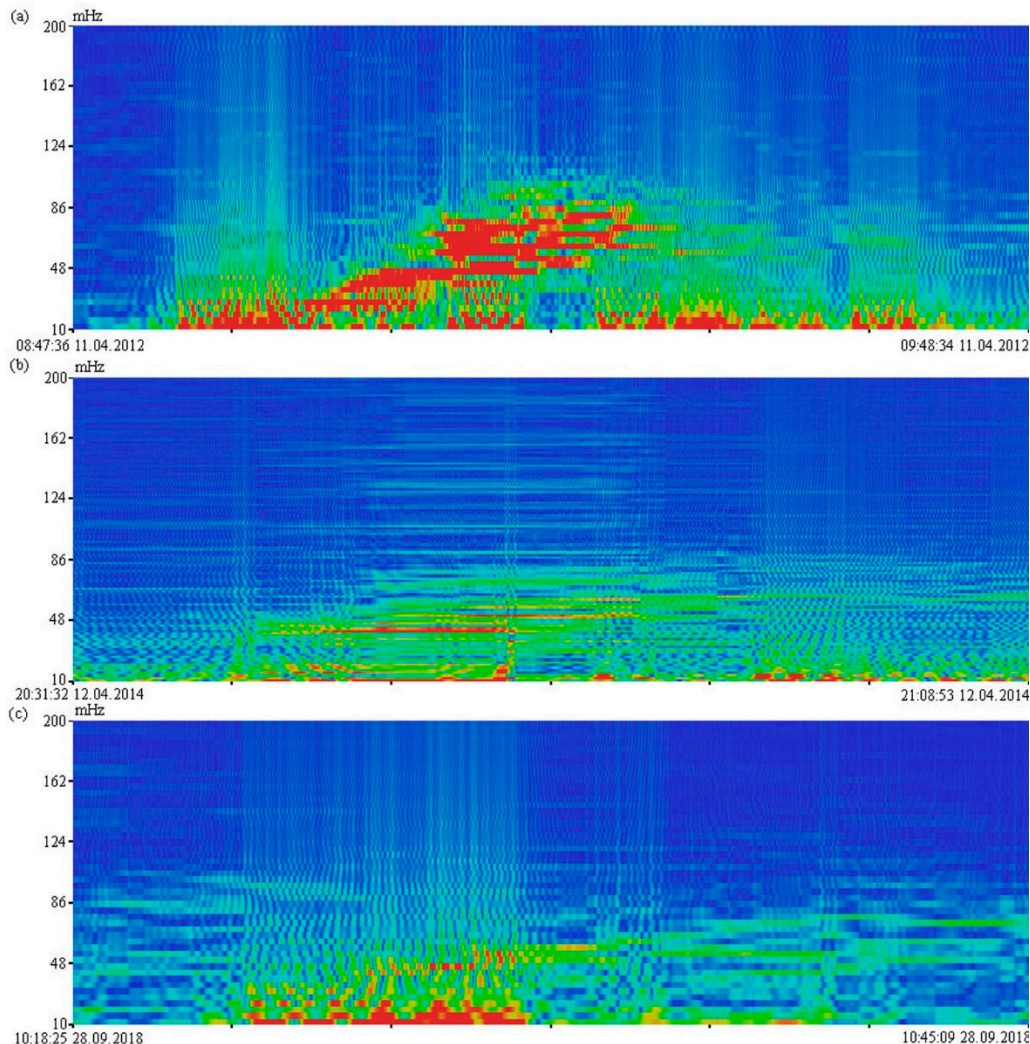

**Figure 4.** Dynamic spectrograms of 52.5 m laser strainmeter recordings from April 2012 (**a**), April 2014 (**b**), and September 2018 (**c**).

In the dynamic spectrograms of the laser strainmeter recording (Figure 4b), we can identify the signal of the earthquake, which occurred on 12 April 2014 at 20:14:39 at 11.270° S, 162.148° E, near the Solomon Islands, at a depth of 22.6 km and a magnitude of 7.6. In the coastal zone, the tsunami height reached 0.5 m. The laser strainmeter, located at a distance of over 6700 km, recorded the earthquake signal almost 20 min later at 20:33:58. For this earthquake, the average speed was 5.58 km/s. In the dynamic spectrogram of the laser strainmeter recording shown in Figure 4b, the earthquake signal amplitude is lower than in the previous case, but it also contains oscillations with periods ranging from 30 to 14 s.

The next earthquake under study occurred on 28 September 2018 at 10:02:45, with a magnitude of 7.5 and wave height of about 11 m. The earthquake epicenter was located at 0.256° S, 119.846° E, at a depth of 20 km and a distance of more than 4800 km from the laser strainmeter. The calculated average speed of the surface elastic wave was 5.49 km/s. On the dynamic spectrogram of the laser strainmeter record (Figure 4c), the earthquake signal was recorded 15 min later at 10:17:19. In the spectrogram, there were oscillations at periods of about 20 s peculiar to earthquakes of such magnitude. From the analysis of the dynamic spectrograms of the three earthquakes that occurred in Indonesia, it follows that along with the oscillations of the earthquake itself, which simply "shake" the Earth, there are disturbances in the lower frequency range.

Figure 5 shows the fragments of the laser strainmeter recordings at the times of registration of the three earthquakes in Indonesia. All figures show a deformation jump peculiar to tsunamigenic earthquakes. For example, in Figure 5a–c, the middle line of the direction of the laser strainmeter recording in the absence of a jump is marked in red, but at the time of the earthquake the recording deviated from its natural behavior (a deformation jump was observed), indicating the tsunamigenic nature of the earthquake.

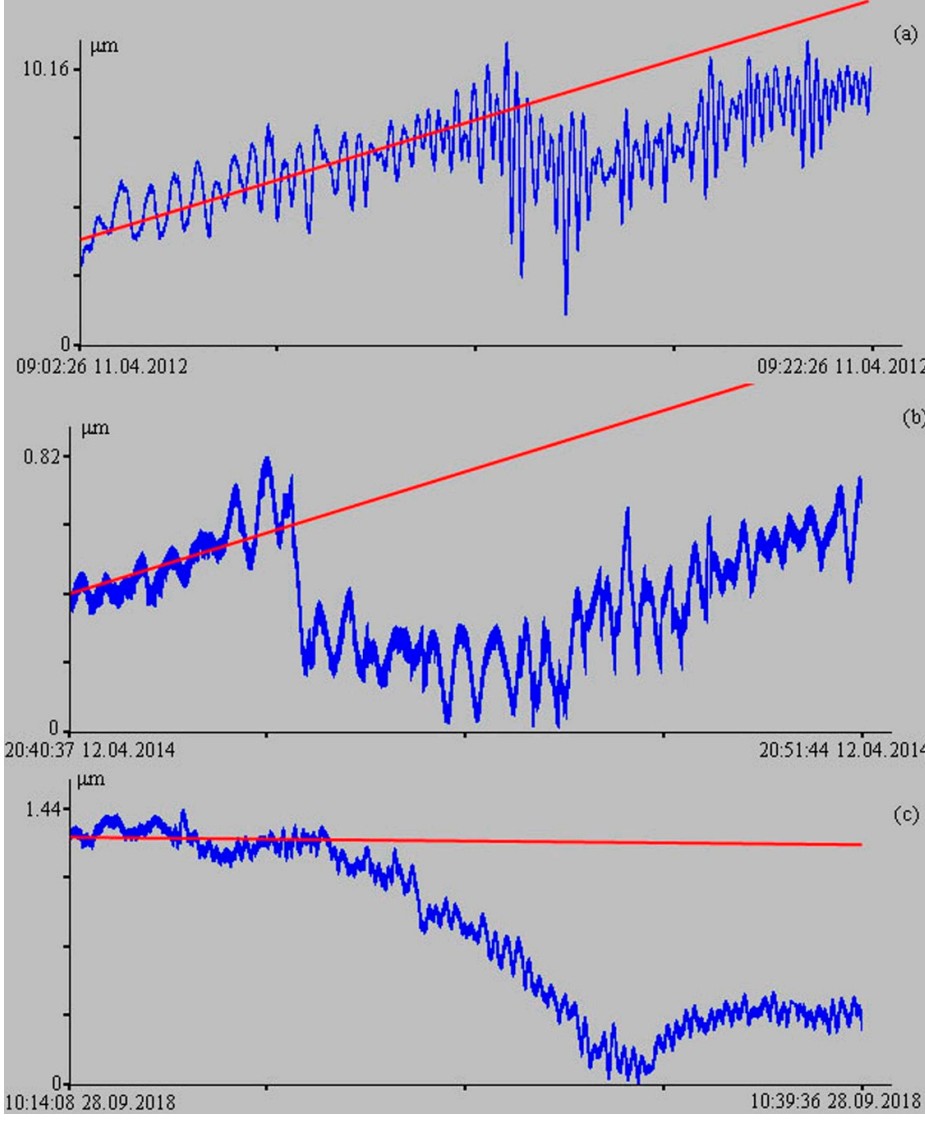

**Figure 5.** Fragments of 52.5 m laser strainmeter recordings from April 2012 (**a**), April 2014 (**b**), and September 2018 (**c**).

### 3.2. Earthquakes in Chile

The recordings from the 52.5 m laser strainmeter showed that three strong earthquakes occurred off the coast of Chile from 2010 to 2018. The first earthquake occurred on 27 February 2010 at 06:34:11 on the northwest coast of Chile at 36.122° S, 72.898° W, at a depth of 22.9 km; the maximum height of the catastrophic tsunami was 29 m. The distance from the earthquake epicenter to the location of the laser strainmeter was more than 17,800 km. The 52.5 m laser strainmeter recorded the signal of this earthquake at 07:19:00. Let us calculate the average speed of propagation of the elastic wave, which is equal to 6.77 km/s. When analyzing the dynamic spectrogram (Figure 6a) of the recording during this earthquake, we identified not only oscillations typical for an earthquake, but also a disturbance in the lower frequency range.

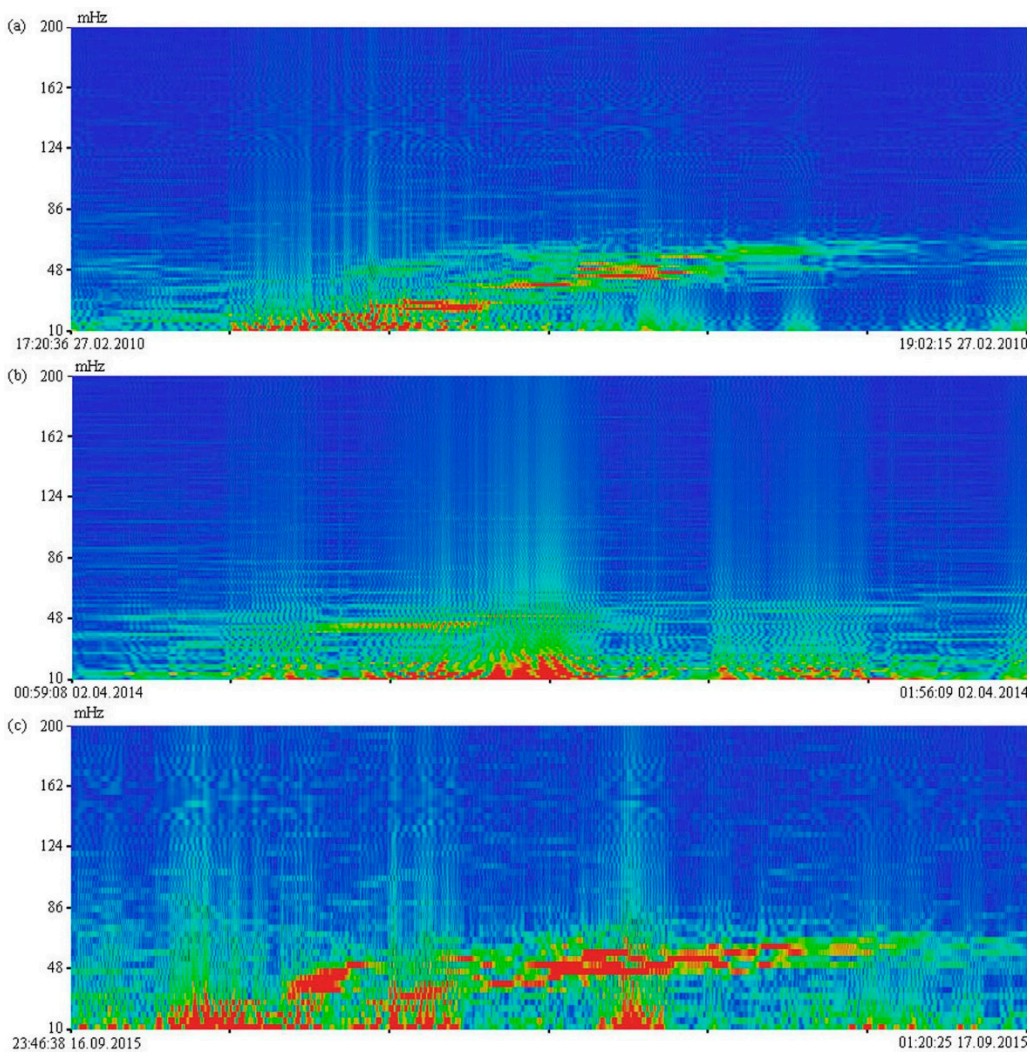

**Figure 6.** Dynamic spectrograms of 52.5 m laser strainmeter recordings from February 2010 (**a**), April 2014 (**b**), and September 2015 (**c**).

Let us analyze the dynamic spectrograms of the fragments of the 52.5 m laser strainmeter records from April 2014 and September 2015. During this period, there were two strong tsunamigenic earthquakes off the northwest coast of Chile. On 1 April 2014, at 23:46:47, a strong earthquake occurred at 19.610° S, 70.769° W, at a depth of 25 km, with a wave height of 4.6 m near the coast. The signal for this earthquake was recorded by the laser strainmeter at a distance of more than 16,700 km on 2 April 2014 at 00:24:10. On 16 September 2015 at 22:54:32, a strong earthquake occurred, with its epicenter at 31.573° S, 71.674° W, at a depth of 22.4 km. As a result of the earthquake, a tsunami with a height

of 13.6 m was generated. The laser strainmeter located at a distance of about 17,650 km recorded the signal for this earthquake at 23:45:01. For these earthquakes, the average speeds of elastic wave propagation were 7.44 km/s and 6.47 km/s, respectively. In the dynamic spectrograms of the laser strainmeter recordings of these earthquakes (Figure 6b,c), oscillations occurred at periods of about 20 s, which are peculiar to earthquakes of this magnitude. Moreover, we noted disturbances in the lower frequency range.

When analyzing the recordings from the laser strainmeter at the times these earthquakes occurred, we identified deformation jumps. Figure 7a–c shows fragments of these earthquakes recordings, whereby the red line indicates the medium direction of the laser strainmeter recording and the deviation from this line at the time the seismic waves were registered indicates the tsunamigenic nature of the earthquakes (the presence of a deformation anomaly—a deformation jump).

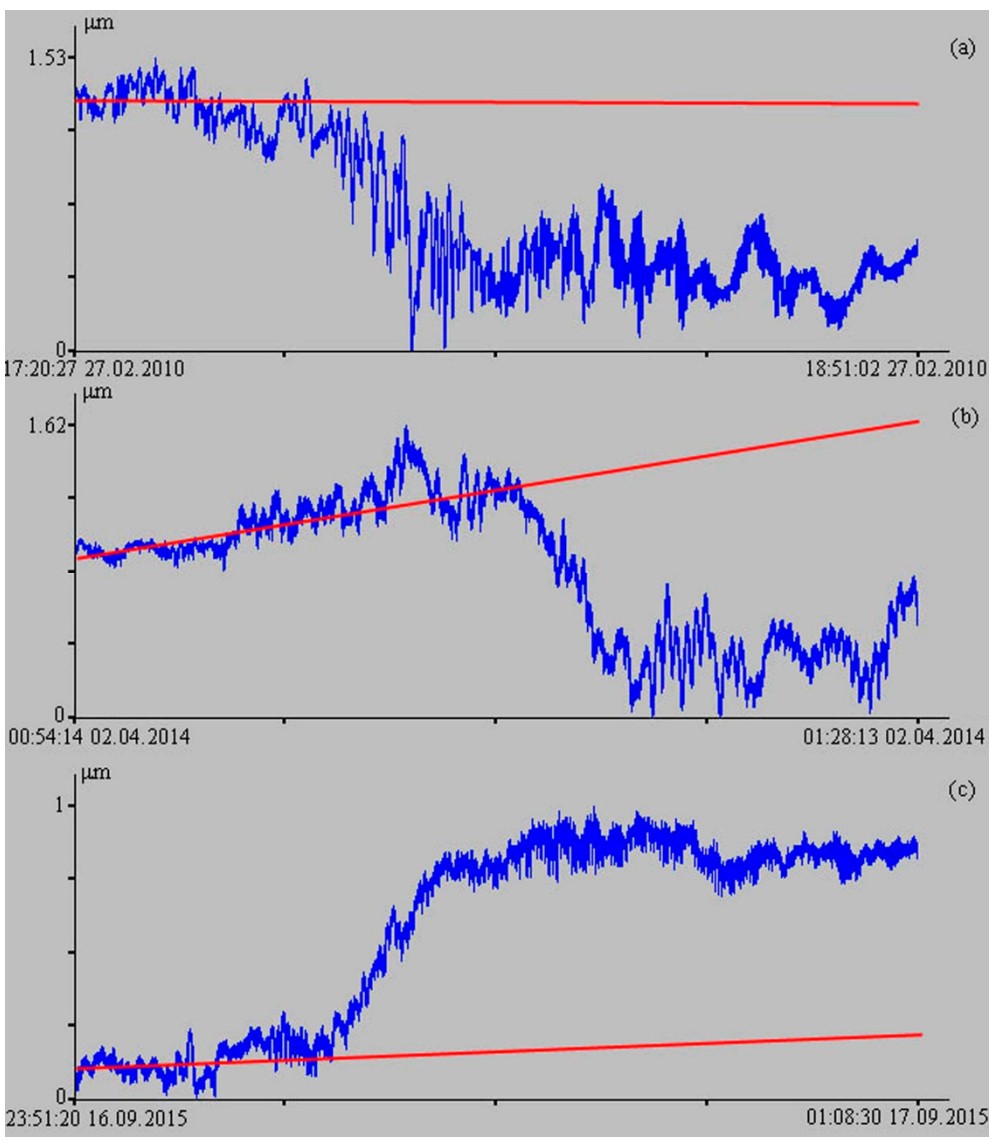

**Figure 7.** Fragments of 52.5 m laser strainmeter recordings from February 2010 (**a**), April 2014 (**b**), and September 2015 (**c**).

### 3.3. Earthquakes on the West Coast of North America

The first powerful earthquake occurred on 28 October 2012 at 3:04:08 on the southwest coast of Canada at 52.788° N, 132.101° W, at a depth of 14 km, with a magnitude of 7.8 and a tsunami height of 12.98 m on the shelf. The distance from the earthquake epicenter to

the laser strainmeter was almost 6800 km. The laser strainmeter recorded this earthquake signal almost 19 min later at 03:23:13. For this earthquake, the average propagation speed of the elastic wave was 5.94 km/s. In the dynamic spectrogram (Figure 8a), we can see the oscillations ranging from 30 to 14 s, which are peculiar to earthquakes of this magnitude, as well as a strong disturbance in the lower frequency range.

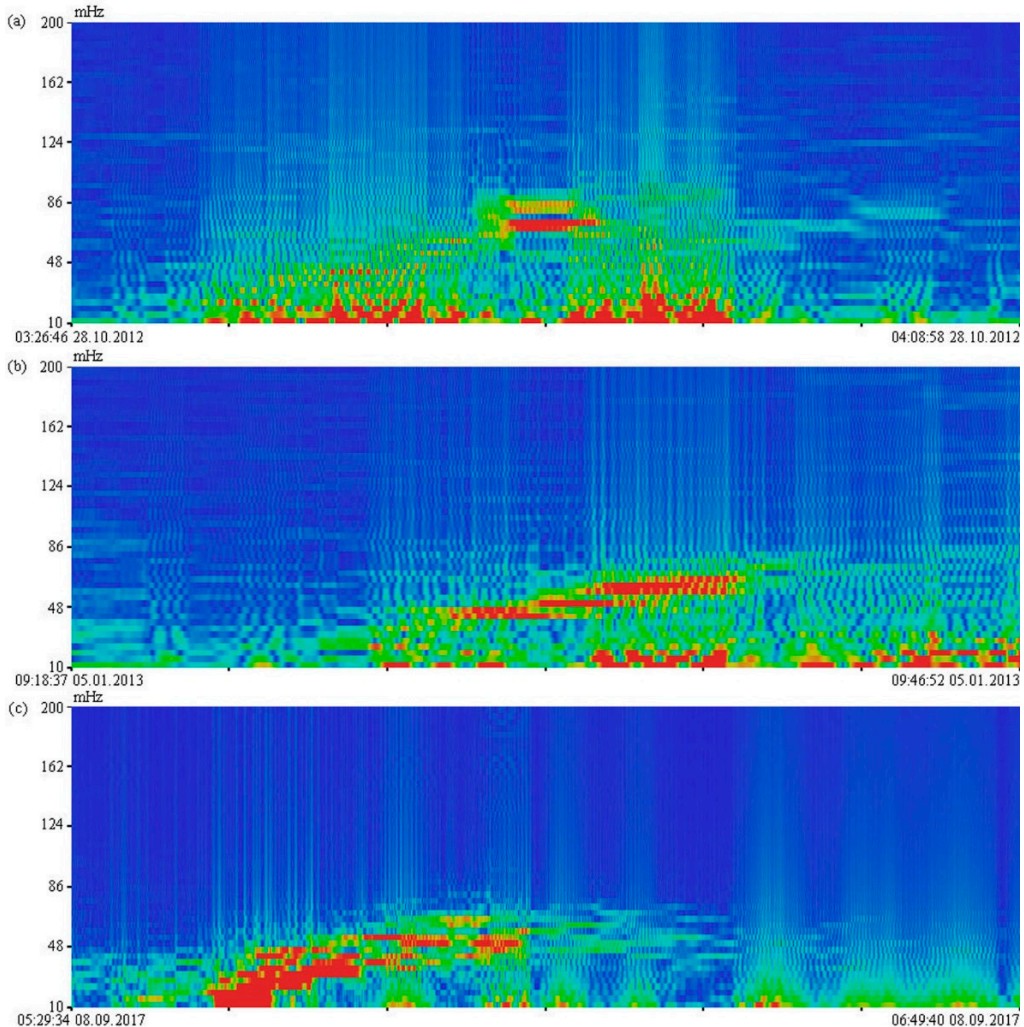

**Figure 8.** Dynamic spectrograms of 52.5 m laser strainmeter recordings from October 2012 (**a**), January 2013 (**b**), and September 2017 (**c**).

Figure 8b shows the dynamic spectrogram of the fragment of the laser strainmeter recording for 5 January 2013, in which we can identify the tsunamigenic earthquake that occurred at 8:58:14 off the coast of Alaska, USA. The earthquake, with a magnitude of 7.5, occurred at 55.228° N, 134.859° W, at a depth of 8.7 km, resulting in a tsunami with a maximum height of 1.5 m. The signal for this earthquake was detected in the laser strainmeter recordings at 09:16:31. The laser strainmeter was located 6500 km away from the epicenter. In the dynamic spectrogram of the laser strainmeter recording (Figure 8b), oscillations occurred from 30 to 14 s and disturbances occurred in the lower frequency region. Another earthquake occurred off the coast of Mexico on 9 August 2017 at 4:49:19, with a magnitude of 8.2. After this, a tsunami with a height of 2.7 m appeared. The earthquake occurred at 15.022° N, 93.899° W, at a depth of 47.4 km, at a distance of 12,150 km from the laser strainmeter. Let us calculate the average propagation speed of the elastic waves. For the 2013 earthquake the speed was equal to 5.92 km/s, while for the 2017 earthquake the speed was 5.48 km/s. In the dynamic spectrogram of the laser strainmeter record (Figure 8c), the earthquake signal was detected at 05:34:28. In the spectrogram,

along with oscillations from the earthquake at periods ranging from 30 to 14 s, disturbances in the lower frequency range also occurred.

Figure 9a–c shows fragments of the laser strainmeter recordings at the times the three earthquakes occurred off the west coast of North America. All figures show deformation jumps peculiar to tsunamigenic earthquakes. In the figure, the middle line of the laser strainmeter recording direction in the absence of a jump is indicated in red, although at the times the earthquakes occurred, the recordings deviated from their natural behavior, showing the tsunamigenic nature of the earthquakes.

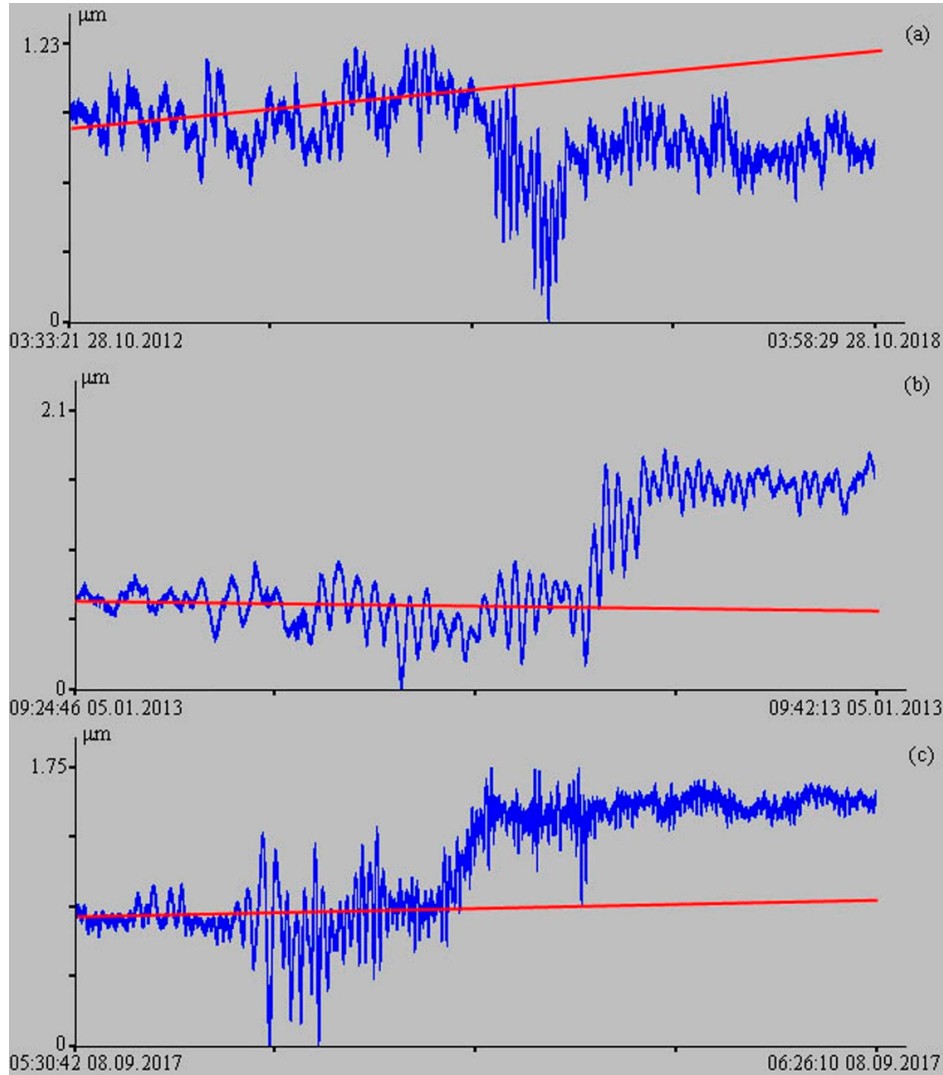

**Figure 9.** Fragments of 52.5 m laser strainmeter recordings from October 2012 (**a**), January 2013 (**b**), and September 2017 (**c**).

Analysis of the dynamic spectrograms of the laser strainmeter recordings of all earthquakes showed that along with the oscillations of the earthquakes themselves, with periods ranging from 30 to 14 s, there were also disturbances in the lower frequency range.

## 4. Analysis of Certain Characteristics of the Registered Deformation Anomalies

An earthquake that occurs near a geological fault provokes the displacement of geoblocks relative to each other. It may also break the connections between geoblocks and cause destruction of separate geoblocks, with powerful movement of the released individual geoblocks and their parts. It is this displacement, along with landslides, that is the cause of a tsunami. It is impossible to register this shift directly in the center; it can only

be determined remotely. At large distances, the amplitudes of these slow displacements are very small. No instruments used in tsunami warning services are capable of registering such displacement; therefore, we can estimate this instead with various models, using continuous geodesic survey data, data from GPS receivers with low sampling rates, and tsunami data produced by the DART network. Models of finite faults from the USGS NEIC have an advantage. They use a kinematic approach based on the method proposed by Ji [19]. For calculations, both the body waves P and S and the Rayleigh and Love surface waves are used. To estimate the dissipative characteristics of deformation anomalies recorded by the 52.5 m unequal-arm laser strainmeter, we will use the calculated displacements of geoblocks (plates, parts of geoblocks, etc.) in the earthquake center according to this model. Table 1 lists the calculated displacements in the earthquake center and deformation anomaly values that the laser strainmeter registered at the moments the earthquakes were recorded.

**Table 1.** Calculated displacements of geoblocks and deformation anomaly values recorded by the laser strainmeter.

| Date | Location | Calculated Displacement, m | Displacement on the Strainmeter, μm |
|---|---|---|---|
| 27 February 2010 | Chile | 10.5 | 1.11 |
| 11 April 2012 | Indonesia | 5.4 | 2 |
| 28 October 2012 | Canada | 1.5 | 0.4 |
| 5 January 2013 | USA | 3 | 0.8 |
| 1 April 2014 | Chile | 8 | 1 |
| 12 April 2014 | Solomon Islands | 0.8 | 0.4 |
| 16 September 2015 | Chile | 3.2 | 0.6 |
| 8 September 2017 | Mexico | 4 | 0.5 |
| 28 September 2018 | Indonesia | 1.8 | 1 |

Since the intensity is proportional to the square of the amplitude, damping of the oscillation amplitude will be expressed by the law of intensity damping, whereby only the damping ratio will be two times smaller. To calculate the damping ratios, we will use the formula used to calculate the oscillation amplitude at the distance under consideration [20]:

$$A = A_0 e^{-\frac{1}{2}\mu x} \tag{1}$$

where $A$ is the amplitude at the registration site, $A_0$ is the initial amplitude, $\mu$ is the damping ratio, and $x$ is the distance.

Let us calculate the damping ratios for each earthquake using the displacement data from Table 1 and the distance from the earthquake epicenter to the laser strainmeter location.

From the obtained results (Table 2), it follows that the damping ratios for all considered tsunamigenic earthquakes are approximately identical. The average damping ratio for all earthquakes was 0.03.

**Table 2.** Earthquake damping ratios.

| Date | Location | Damping Ratio, km$^{-1}$ |
|---|---|---|
| 27 February 2010 | Chile | 0.033 |
| 11 April 2012 | Indonesia | 0.026 |
| 28 October 2012 | Canada | 0.032 |
| 5 January 2013 | USA | 0.032 |
| 1 April 2014 | Chile | 0.032 |
| 12 April 2014 | Solomon Islands | 0.027 |
| 16 September 2015 | Chile | 0.026 |
| 8 September 2017 | Mexico | 0.034 |
| 28 September 2018 | Indonesia | 0.027 |

## 5. Discussion

Let us note some of the peculiarities mentioned above: (1) tsunamigenic earthquakes are characterized by the presence of a deformation anomaly—a deformation jump—and disturbances in the dynamic spectrogram in the lower frequency range, in comparison with the range of oscillations, which is peculiar to the earthquake source zone; (2) the damping ratios of deformation anomalies for all regions of the Earth are the same, within the measurement and calculation error ranges; (3) the durations of deformation anomalies for different tsunamigenic earthquakes vary from 15 s to 17 min.

Let us analyze some of the peculiarities of the above-mentioned tsunamigenic earthquakes. Let us focus on the presence in the dynamic spectrograms of disturbances in the low-frequency and ultra-low-frequency ranges that are not associated with oscillations, which are excited in the source zones of earthquakes. These disturbances are associated not with natural processes, but with the processing of records containing deformation jumps. The appearance of disturbances indicates only one thing, namely the presence of a deformation anomaly—a deformation jump—in the record. There are no oscillations in these areas. Let us demonstrate this using the example of processing an instrument recording containing oscillations caused by sea wind waves and a jump. Figure 10a shows an instrument recording containing a jump. Figure 10b shows its spectrum. Figure 10c shows the same instrument recording but without a jump, while Figure 10d shows its spectrum. All scientists involved in signal processing understand this effect. The increase in intensity in the low-frequency range is associated not only with the Gibbs phenomenon, but mainly with the presence of a jump in the recording; thus, the presence of disturbances in the spectrograms of tsunamigenic earthquake recordings containing deformation anomalies indicates only one thing—the influence of this deformation jump on the increases in intensity in the low-frequency and ultra-low-frequency ranges due to the processing effect, which is also remarkable. After all, looking only at the spectrogram, which was obtained in real time, one can notice the tsunamigenic nature of the earthquakes.

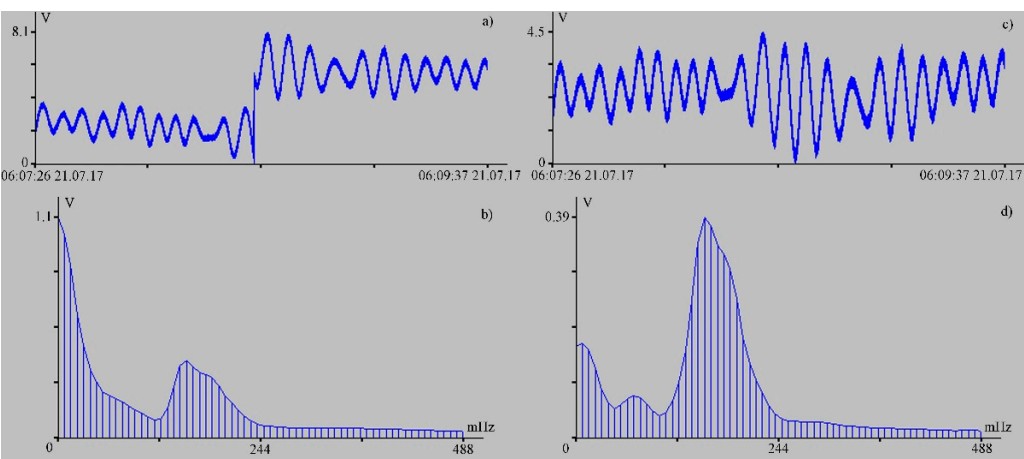

**Figure 10.** Fragment of the laser strainmeter recording showing hydrosphere pressure variations [21] when registering surface wind waves (**a**), along with its spectrum (**b**). The same recording is shown with the jump removed (**c**), along with its spectrum (**d**).

Let us pay some attention to the fact that for different regions of the Earth, the damping ratios of deformation anomalies are practically the same. This coefficient should consist of summands associated with divergence and absorption due to dissipative energy losses. If we consider only the cylindrical divergence, then the signal amplitude should decrease with distance in proportion to the square root of the distance. Regarding spherical divergence, the signal amplitude, with distance, should decrease in proportion to the distance. Solving the inverse problem, using Table 1, on the basis of the data from the laser strainmeter, for the spherical divergence we can obtain the following initial amplitudes of deformation

anomalies (deformation jumps), which arose at the source of tsunami generation. In the sequence of column 3 in Table 1 (calculated displacement), these values are 19.6 (10.5 m), 16.7 (8 m), 10.6 (3.2 m), 11.6 (5.4 m), 2.7 (0.8 m), 4.8 (1.8 m), 2.7 (1.5 m), 5.2 (3 m), and 6.0 m (4 m). Taking into account the fact that the calculated data almost coincide with the model data (given in parentheses), we can state that the deformation anomaly arising at the source of tsunami generation, in the case of spherical divergence, moves similarly to the motion of a soliton; however, all calculations are correct when the signal moves along the surface of the Earth, i.e., in an arc rather than a chord. We do not know the path of the signal, so we take the limiting case, whereby the distances from the place of generation to the place of registration are equal to the lengths of the arcs of the circles, as determined by the coordinates of the points. Let us discuss the probable physical mechanism of formation of the deformation anomalies (deformation jumps) arising during the movement of geoblocks (joints) of the Earth's crust at the source of a tsunami generation. Let us assume that when an earthquake occurs, geoblocks are displaced relative to each other or one of the geoblocks (geological plate, joint of the Earth's crust) becomes out of balance and moves relative to the other geoblocks. This movement leads to the movement of huge masses of water, which subsequently degenerate into a tsunami. We are not interested in the origin point of the tsunami. We are only interested in the movement of the geoblock or geoblocks relative to each other. We can describe these geoblocks movements using the following equation:

$$\frac{\partial^2 u}{\partial t^2} + \sin u - \frac{\partial^2 u}{\partial x^2} = 0 \tag{2}$$

where $u$ is the displacement of a geoblock. This is the classic sine-Gordon equation. One of the solutions for this equation in the factorized form, which is peculiar to solitons, allows us to obtain the following geoblock displacement:

$$u = 4arctg\left[\exp\left(\pm\frac{x - Vt}{\sqrt{1 - V^2}}\right)\right] \tag{3}$$

where $V = 0.5$. A solution with "+" gives a kink, while a solution with "−" gives an anti-kink. Figure 11 shows the displacement of the geoblock in the form of an anti-kink.

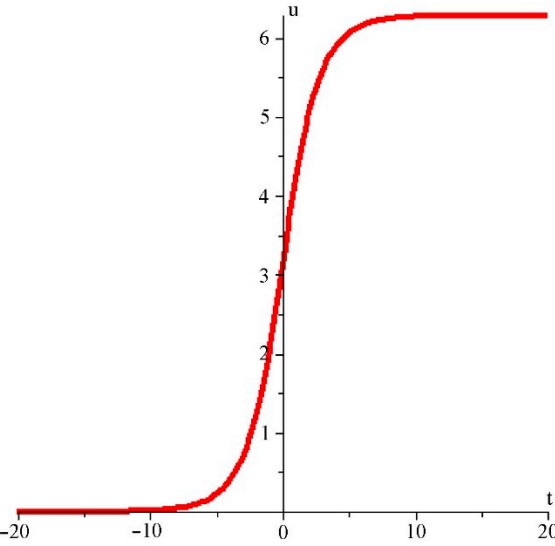

**Figure 11.** Displacement $u$ at $V = 0.5$, shown as a function of time (anti-kink).

The displacement as a function of the coordinate and time at $V = 0.5$, demonstrating plastic deformation, is shown in Figure 12.

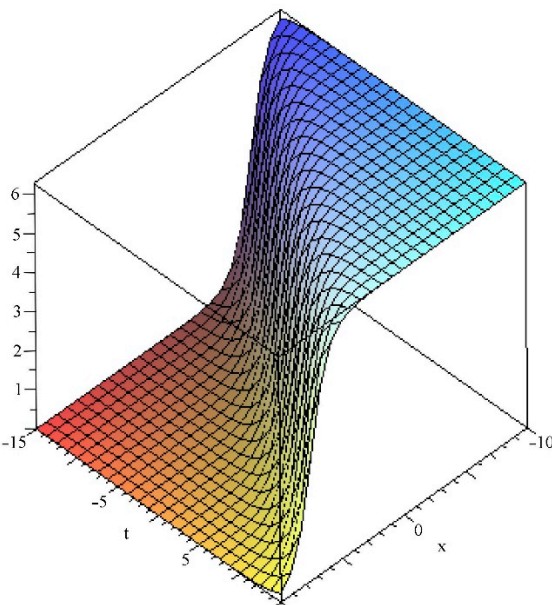

**Figure 12.** Geoblock displacement in the case of plastic deformation.

These geoblock displacements, when interacting with the environment, are transmitted to the surrounding space and propagate in the Earth in the form of a soliton—a deformation step, corresponding to a kink or anti-kink.

In addition to the exponential function, hyperbolic functions also satisfy Equation (2). In this case, the solution of Equation (2) gives a two-soliton solution. For this case, the displacement of the geoblock (homogeneity of the Earth's crust, a plate, etc.) depending on the coordinate and time for $V = 0.5$ is shown in Figure 13. This corresponds to elastic deformation under stretching. Plastic deformation can also occur under stretching only in the vicinity of $x = 0$.

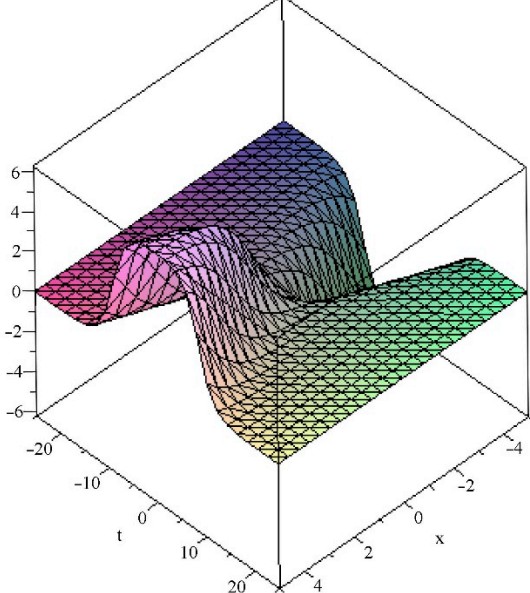

**Figure 13.** Geoblock displacement in the case of elastic deformation.

Under special conditions, such as for imaginary solutions, the solution to Equation (2) will give breathers. Figure 14 shows the geoblock displacement in the breather at $V = 0.5$.

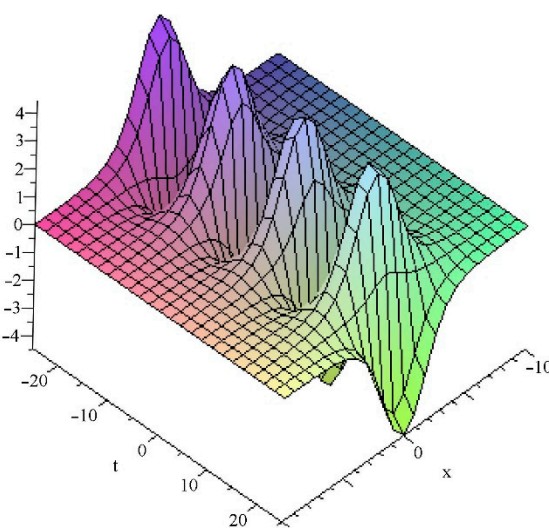

**Figure 14.** Geoblock displacement in the breather at *V* = 0.5.

These displacements (several deformation jumps) were observed during the 2011 Great East Earthquake in Japan (Tohoku). It is clear that any movements of a geoblock (or geoblocks) are transmitted to the neighboring environment, in which deformation anomalies propagate in the form of kinks, anti-kinks, and other similar disturbances.

Next, let us consider the behavior of an underwater landslide. At the advanced stage of the process, an underwater landslide can be represented as consisting of two parts: the head part (front) is a soliton (kink), while the tail part is a periodic wave. The landslide front is a same kink (bore) and can be described by a single-kink solution of the sine-Gordon equation, with "+" on the right-hand side of Equation (3). The formed soliton moves without experiencing any resistance from the medium. In expanding areas, the landslide front begins to blur. In such cases, the height of the soliton begins to decrease due to the energy conservation law. This process can be schematically represented in the following sequence during formation of an underwater landslide: (1) in the initial stage, a soliton (kink) and a periodic part are formed; (2) leaving the source of origin, the soliton begins to move in its environment with conservation of energy; (3) the motion of the soliton in the environment obeys the law of motion, ranging from cylindrical to spherical divergences. Taking into account the above, we must register a soliton with an ever-decreasing height and with increases in distance from the place of its registration. Figure 5 shows a laser strainmeter recording containing a deformation disturbance (a deformation jump) kink caused by an underwater landslide during the earthquake in Indonesia. Both earthquakes and underwater landslides can form the periodic oscillations observed in the figure.

## 6. Conclusions

During the processing of the experimental data from the laser strainmeter, we found that all tsunamigenic earthquakes are characterized by the presence of deformation anomalies—deformation jumps—in the instrument records. These deformation anomalies, leading to the formation of tsunamis in the vicinity of the earthquake source areas, occur during the relative movement of geoblocks (plates, joints) and underwater landslides. These geoblock movements can be described by the sine-Gordon equation, the one-kink and two-kink solutions of which explain the appearance of the observed deformation anomalies in laser strainmeter recordings. The behavior of deformation anomalies is the same as the behavior of solitons in non-linear medium. Considering that the signals from tsunamigenic earthquakes containing deformation anomalies propagate at speeds much higher than those of surface waves (from 5.48 to 7.44 km/s), we can assume that the signals do not propagate along the Earth's surface, with the divergences ranging from cylindrical to spherical. Further research should investigate the spatial behavior of the deformation anomalies from

tsunamigenic earthquakes. For this, it will be necessary to place several laser strainmeters far from each other, along the assumed direction of movement of deformation anomalies (deformation jumps), kinks, anti-kinks, and breathers. These experimental studies will allow us to study the main parameters of the observed disturbances. Of particular interest is the conservation of the soliton shape with decreasing value due to divergence in space during movement. The development of this area of research, along with the application of the classical "magnitude geographical principle" used for determining the tsunami hazard of underwater earthquakes will bring us closer to short-term tsunami forecasting. Taking into account the above, with spherical divergence in accordance with Table 1, we can calculate the applicability of GPS receivers capable of registering displacement with an accuracy level of 1 mm for recording displacements (column 3 of Table 1). In this way, with an average displacement of 4.2 m (according to Table 1), a GPS receiver will be able to register a displacement of 2 mm at a distance of 2100 m under conditions of spherical divergence. It is clear that there is no point in discussing the prospects of using GPS receivers for registering displacements of geoblocks (plates, joints) leading to the occurrence of tsunamis. The cylindrical divergence of the signal does not help in situations.

**Author Contributions:** G.D.—problem statement, discussion, and writing of the article. S.D.—data processing, discussion, and writing of the article. All authors have read and agreed to the published version of the manuscript.

**Funding:** The work was carried out with financial support from the Russian Federation represented by the Ministry of Science and Higher Education of the Russian Federation, Agreement No. 075-15-2020-776.

**Data Availability Statement:** 3rd Party Data. Restrictions apply to the availability of these data.

**Acknowledgments:** We would like to express our deep gratitude to all employees of the Physics of Geospheres laboratory.

**Conflicts of Interest:** The authors declare no conflict of interest.

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
