# Peer review of "Deformation Anomalies Accompanying Tsunami Origination"

_jmse, doi:10.3390/jmse9101144_

Round 1
Reviewer 1 Report
I insist my original judgement, which is that the observations are interesting, but the interpretation is not convincing. Earthquakes that do not generate tsunamis also cause Earth deformation, i.e., geoblock movements as mentioned by the authors. I don't see any theories in your article attempting to discriminate between these two types of earthquakes. I will leave the editor to make the final decision.
Reviewer 2 Report
Dear authors,
The paper is ready for publication.
This manuscript is a resubmission of an earlier submission. The following is a list of the peer review reports and author responses from that submission.
Round 1
Reviewer 1 Report
Summary
This paper reported characteristic deformation anomaly recorded by a laser strainmeter, and proposed a method to estimate magnitude of tsunami by using the ground-surface displacement data. It is an interesting experiment, but which deformation anomaly is important for estimation of tsunami size is unclear. In addition, the method they developed is not fully explained in the manuscript. Therefore, it is difficult to understand not only what author did, but also the validity of the method. This reviewer considers major revisions are required.
Major comments
- One of my main concerns is that it is unclear what "the deformation method" is, although the method is the main topic in this manuscript. The title of section 3 is “description of the deformation method”, but there is no clear description about what the "method" is. Instead of the explanation of the method, in the section authors mentioned the following two characteristic variations recorded by a strainmeter during the earthquakes: deformation jump and oscillation. However, it is unclear whether the both are important or only one of them is important, when distinguishing between tsunamigenic earthquakes and non-tsunamigenic ones. Actually, in sections 4 to 7, the measurements of the oscillations are explained in detail by showing the dynamic spectrograms during several tsunamigenic earthquakes (Figures 4-6) without no explanation about the deformation jump for these earthquakes. In section 8 (i.e. conclusion), however, importance of the deformation jump is suddenly stated in lines 369-371. It is confusing for readers. My suggestion is that authors should clearly state whether deformation jump or oscillation is more important or both for the deformation method they developed (According to description in lines 142-143, I guess authors considers both deformation jump and oscillation are important). If deformation jump is important, the strainmeter records for the Indonesia, Chile, and other earthquakes should be shown in figures, in addition to the spectrograms shown already.
- What physical phenomena do authors consider deformation jump and oscillation to be, respectively? In particular, there is no mention of the jump in the manuscript. I suspect authors interprets the jump as the static strain change due to permanent deformation of the crust associated with coseismic slip. If so, authors need to cite studies that use such static strain change to estimate earthquake size (e.g., Itaba 2018, EPS). On the other hand, regarding to the oscillation, I understood that authors interpreted it as elastic wave (seismic wave). This is because they calculate the propagation speed as elastic wave and interpret it as a seismic wave in line 262. However, the manuscript does not state what elastic wave (i.e. body wave or surface wave) and its propagation path. In the manuscript, the distance between the epicenter and the strainmeter seems to be the great circle distance along the ground surface, and thus I believe that authors assumed surface waves as the cause of the oscillation. If so, it is necessary for authors to check and mention whether or not the propagation velocity they obtained is consistent with the velocity structure of the surface waves that was previously estimated by the seismological studies.
- Lines 330-365: There is little information on seismic-data analysis that authors used. Therefore, I am not able to follow what kind of analysis was conducted, and it is impossible to judge the validity of the analysis results. More detailed information should be provided in the manuscript. The examples are shown as follows: (1) what does "calculated displacement" mean, and where? Is it the average or the maximum amount of dislocation on the fault plane? Or, is it the average or the maximum amount of earth-surface displacement expected from the coseismic dislocation on the fault plane? In addition, what method (or theory) did authors use to calculate displacement from USGS fault model? The information of the analysis should be shown, so that readers can follow the method (e.g. physical theory, numerical model, and so on. If authors used the method that had been previously proposed by another study, authors should mention it in the manuscript). (2) What is the ”displacement on the strainmeter” shown in the right column of Table 1? Is it read from the oscillation? If so, is it maximum values of both amplitudes, those of half amplitude, or other? (3) Related to comment (1), does "earthquake center" in line 343 indicate the hypocenter (or starting point of coseismic rupture), the centroid, or the geometric center of the fault model used in the calculation? (4) In line 349, the word "damping ratio" appears without any explanation. It is unclear what "damp" means. The equation in line 352 seems to be an expression for intrinsic attenuation, in which seismic wave (elastic wave) energy is converted into thermal energy and dissipated, and so I guess that "damping" means intrinsic attenuation, but this is not obvious. Citing the literature [35] is not enough; the minimum necessary information should be described in the manuscript, so that the reader can understand what physical model was used. (5) According to the description in the manuscript, I guess that authors estimated the value of the damping ratio by assuming the ratio is constant over the path of seismic wave propagation. Is it reasonable assumption? If so, what is the reason for that? In addition, is the obtained value consistent with the intrinsic attenuation (quality factor, Q) of the Earth's interior that is estimated by seismic tomography studies? (6) The strainmeter record and the spectrogram for the 2004 Sumatra earthquake are shown in Figure 2 and the detail of the characteristics is mentioned in the main body, whereas the record for this event was excluded when authors estimate the damping ratio, resulting in no existence in Tables 1 and 2. Why? In addition, in the whole manuscript, why is there no mention of the 2011 Tohoku earthquake, whose source area is located close to the strainmeter? I expect that substantial deformation signal due to the 2011 event was recorded by the strainmeter.
- The purpose of authors’ study is unclear. In lines 10-11, authors mentioned "early warning of possibility of a catastrophic wave occurrence long before it reaches a coast". Which is the target of forecasting, near-field tsunami or far-field tsunami?
- It is unclear that the position of this study in comparison to previous studies such as seismic, GNSS and tsunami studies. Authors reviewed those previous studies in lines 21-131, but did not mentioned the relationship between those studies and authors’ study. Please clarify it.
- In line 374, authors mentioned tsunami height estimation, but there is no explanation about how to predict the tsunami height in the manuscript. The tsunami height along the coasts cannot be estimated from displacement at the source area alone, because the size of a tsunami depends not only on the initial tsunami height, but also the extent of the tsunami source area. I think this feature is already mentioned in line 93-96 by authors themselves. I understood that authors’ method may provide displacement information, but cannot provide source-extent information. Please clarify what information the authors’ method can provide to tsunami forecasting.
- Authors showed Figure 3 to mention that the spectrum amplitude is smaller in the longer period than that around 16 s, which is one of the features for non-tsunamigenic earthquakes. However, the same feature seems to be observed in 2014 Indonesia, one of the tsunamigenic earthquakes shown in Fig. 4 (middle), doesn’t it? Please clarify it.
Minor comments
- Lines 42-63: This comment is related to comment 5. Authors referred Titov et al. to mention a strategy of tsunami early warning: the first tsunami warning will be provided using the GNSS data, and then the warning will be updated by using the tsunami data such as the DART pressure data. However, such strategies had been already proposed by several studies (Melgar and Bock, 2013; Wei et al., 2014; Tsushima et al., 2014). Authors should mention these previous studies. In addition, it is important to note that most of tsunami early warning systems that have been already operated are based on seismic-wave data, especially for the first bulletin. Authors should mention such approaches. For example, the Pacific Tsunami Warning Center (PTWC) uses seismic wave data with long wave period (W-phase) to provide global tsunami warnings to coastal regions along the Pacific Rim (PTWC, 2014). Another example is that the Japan Meteorological Agency (JMA), Japan, provide local tsunami warning within 3 minutes of near-field earthquakes using seismic data (Tatehata, 1997), and then update the warning using seismic-waveform data and tsunami data (Kamigaichi, 2015). In particular, a method to rapidly estimate moment magnitude using seismic-waveform data is already established by Kanamori and Rivera (2008). W-phase appears in the seismic records between P-wave and S-wave arrivals, and can be used to estimate seismic moment, centroid location and fault mechanism. The effectiveness of the W-phase inversion has been already demonstrated by many papers (e.g. Kanamoti and Rivera, 2008). For example, the retrospective analysis for 2011 Tohoku earthquake showed that magnitude 9.0 could be obtained by the W-phase inversion 20 min after the earthquake (Duputel et al., 2011). The W-phase inversion approach has been used operationally by the warning centers such as PTWC and JMA (PTWC, 2014; Kamigaichi 2015). These recent advances to rapidly estimate seismic moment is strongly related to this manuscript, and should be shown in the introduction section.
- Lines 67-68: Authors mention the following: "Most of the current warnings, which include tsunamis, are aimed at 67 the mid-to-far zone regions." Which warning system do authors indicate? References should be shown in the manuscript.
- Line 332: Authors may want to mention dislocation (or slip) on the earthquake-fault plane by using the term "landslide". However, "landslide" implies a phenomenon that is caused by a single force, and it is confusing to use the term "landslide" for expressing earthquake-fault motion that is double-couple source. I recommend to using "dislocation", instead of "landslide".
References
Duputel, Z., Rivera, L., Kanamori, H., Hayes, G.P., Hirshorn, B. and Weinstein, S. (2011). Real-time W phase inversion during the 2011 off the Pacific coast of Tohoku Earthquake. Earth Planet Sp 63, 5. https://doi.org/10.5047/eps.2011.05.032.
Itaba, S. (2018). Rapid estimation of the moment magnitude of the 2011 Tohoku-Oki earthquake (Mw 9.0) from static strain changes. Earth Planets Space 70, 124 (2018). https://doi.org/10.1186/s40623-018-0894-5
Kamigaichi, O. (2015). Tsunami forecasting and warning. In: Meyers R. A. (eds) Encyclopedia of Complexity and Systems Science. Springer, Berlin, Heidelberg. https://doi.org/10.1007/978-3-642-27737-5_568-3.
Kanamori, H., and Rivera, L. (2008). Source inversion of W phase: speeding up seismic tsunami warning. Geophysical Journal International, 175(1), 222–238. https://doi.org/10.1111/j.1365-246X.2008.03887.x.
Melgar, D., and Bock, Y. (2013). Near-field tsunami models with rapid earthquake source inversions from land- and ocean-based observations: The potential for forecast and warning. Journal of Geophysical Research: Solid Earth, 118(11), 5939–5955. https://doi.org/10.1002/2013JB010506.
Pacific Tsunami Warning Center/International Tsunami Information Center (PTWC/ITIC) (2014). User’s guide for the Pacific Tsunami Warning Center enhanced products for the Pacific tsunami warning system. Revised Edition. IOC Technical Series, 105, UNESCO/IOC, Paris, France.
Tatehata, H. (1997). The New Tsunami Warning System of the Japan Meteorological Agency. In: Hebenstreit G. (eds) Perspectives on Tsunami Hazard Reduction. Advances in Natural and Technological Hazards Research, vol 9. Springer, Dordrecht. https://doi.org/10.1007/978-94-015-8859-1_12.
Tsushima, H., Hino, R., Ohta, Y., Iinuma, T., and Miura, S. (2014), tFISH/RAPiD: Rapid improvement of near-field tsunami forecasting based on offshore tsunami data by incorporating onshore GNSS data. Geophysical Research Letters, 41, 3390–3397. https://doi.org/10.1002/2014GL059863.
Wei, Y., Newman, A. V., Hayes, G. P., Titov, V. V., and Tang, L. (2014). Tsunami forecast by joint inversion of real-time tsunami waveforms and seismic or GPS Data: Application to the Tohoku 2011 tsunami. Pure and Applied Geophysics, 171, 3281–3305. https://doi.org/10.1007/s00024-014-0777-z.
Author Response
Hello.
Thank you for the review. The attached file contains the response to the review.

Reviewer 2 Report
This paper argues that using a strainmeter installed in the far field to record Earth surface deformation, it is possible to confirm if an earthquake generates a tsunami.
Major Comments:
(1) The idea is fundamentally flawed. Tsunamis are generated primarily by large fault slip in the shallow portion on a fault. Thus it is essential to identify if such slip occurs in an earthquake in order for tsunami warning purposes. Large slip may happen in the deep portion, leading to similar deformation on land but generating no tsunamis. However, it is generally believed that land-based measurements are not able to constrain shallow slip in the ocean. This is why it is still controversial if tsunamis can be predicted only relying on measurements of land deformation. And it is the reason why so many tsunami buoys and seafloor observation networks are installed in the ocean. Thus, even near-field land deformation cannot predict tsunamis solely, it is far from convincing to use far-field data as proposed by the authors.
(2) Link from data to conclusion lacks physics. It is reasonable to think that the deformation anomaly is related to the earthquake. However, why is it due to the permanent deformation of the earthquake? What is the physics?
(3) Data analysis lacks knowledge of earthquake dynamics. The authors keep using a vague term "low-frequency oscillation". What frequency band do you exactly mean? Why is this low-frequency band related to tsunamis? From the figures it seems that the authors mean approximately 80~100s, which seems a very widely-analyzed frequency band of Rayleigh waves. It has nothing to do with tsunamis.
(4) The introduction section is very badly organized. The authors seem to be unfamiliar with earthquakes and tsunami warnings, and cite many papers that are not relevant here. I suggest compress this section and make it more organized and focused on your topic.
Minor Comments:
Figure 2,3: Do not show absolute UTC time. Change the horizontal axis to relative time to the earthquake origin time.
Figure 2: you are saying that the deformation anomaly is related to the earthquake. Why is there not seismic waves in the record, as similar to Figure 3? The anomaly gradually increases -- how is this explained? Earthquake-generated permanent deformation should be formed in an instant.
Line 8: Basing on -> Based on
Line 30: If the problem -> Although the problem
Line 40: in this review -> in this article? I think this is not a review paper.
My conclusion is that this manuscript is not fit for publication. I also cannot see how this manuscript can evolve quickly to an acceptable standard and thus recommend it be rejected. My only suggestion to the authors is that possibly the authors may change the topic to something like "Strain observations from large earthquakes in Russia", and only conduct the data analysis without mentioning the big hope of tsunami warning.
Author Response

(The authors gave the same response as above.)

Round 2
Reviewer 1 Report
The authors mention that deformation jump in strain record is important in discriminating tsunami generation. However, it is unclear what physical phenomenon deformation jump is based on. The other reviewer and I pointed this out in the first review, but the authors have not been revised. Thus, this manuscript is not suitable for publication. I recommend this manuscript to be rejected. I hope that the manuscript will be resubmitted after physical interpretation of the deformation jump is presented.
Reviewer 2 Report
I appreciate the authors' efforts to revise the paper. I have carefully read the revised part. Unfortunately, I find that the authors did not address the essential question. Let me explain the problem in details again, which might be useful for the authors to re-organize the paper.
You argue that a deformation jump on a laser strainmeter relates to the generation of a tsunami. However, through the paper, you did not explain what this signal is. Earthquakes generally cause two kinds of signals -- seismic waves and permanent deformation of the Earth surface. Which kind is the signal in this paper? From the term "deformation jump", and since this signal does not look like a wave, you seem to indicate that it is the permanent deformation. Then there will be two problems you need to clarify:
(1) Permanent deformation means 0 frequency. What is the purpose of calculating the spectrograms? In my original comments I mentioned that you keep using the term "low-frequency oscillation", which seems to mean periods of 80-100s. This frequency range is a typical range of seismic waves, which has nothing to do with permanent deformation. Are you analyzing the seismic waves?
(2) If the deformation jump is permanent deformation, then it goes back to my first comment in the original review, which is, permanent deformation is such a great distance is not reliable for tsunami prediction. There are many studies to show that only using land-based deformation data will lead to large errors in predicting tsunamis. The reason is that land-based deformation depends on the fault slip under land, while tsunamis are mostly caused by fault slip under ocean (or equivalently shallow slip near trench). Thus my original suggestion was to only present your observations. You could have some discussions that such observations could be useful for tsunami warning, but I believe it won't be reliable.
Your attenuation law also does not have physical bases. If the signal is permanent deformation, then there is an analytical solution of the deformation, although the solution only applies to small distances when curvature of the Earth surface is ignored.
There are other statements that sound casual and are not scientific. For instance, Line 409, "From the analysis of the dynamic spectrograms of the three earthquakes that occurred in Indonesia, it follows that along with the oscillations of the earthquake itself, which simply “shake” the Earth, there are disturbances in the lower frequency region, which are the cause of a tsunami." Tsunamis are caused by permanent seafloor deformation, not "lower frequency oscillations". Line 530, "Since the intensity is proportional to the square of the amplitude, damping of the oscillations amplitude will be expressed by the law of intensity damping, only the damping coefficient will be two times smaller." I don't see a reference here except for a textbook of physics. Are you talking about the attenuation of seismic waves? Then there will be many research papers on this topic. Are you talking about the damping of permanent deformation? Then there is the analytical solution you can refer to.
In my original review I suggest the authors suppress the introduction part and only keep those references that are relevant. The authors did not do so. They instead cite more irrelevant papers. My opinion is that you don't have to cite all the papers you have read. And you don't summarize every paper with details. I think this comment is important, but it is not essential for me to insist rejection of the paper.
Overall, I still think this paper is not ready for publication. I suggest the authors either avoid analysis of earthquake dynamics and only present the observations, or collaborate with some earthquake expertises.